# A Compact RF Energy Harvesting Wireless Sensor Node with an Energy Intensity Adaptive Management Algorithm

**DOI:** 10.3390/s23208641

**Published:** 2023-10-23

**Authors:** Xiaoqiang Liu, Mingxue Li, Xinkai Chen, Yiheng Zhao, Liyi Xiao, Yufeng Zhang

**Affiliations:** School of Aeronautics, Harbin Institute of Technology, Harbin 150001, China; 20b921030@stu.hit.edu.cn (X.L.); li_mingxue@hit.edu.cn (M.L.); 15069301390@126.com (X.C.); 22s136086@stu.hit.edu.cn (Y.Z.); xiaoly@hit.edu.cn (L.X.)

**Keywords:** RF energy harvesting, rectenna, compact design, WSN, energy management

## Abstract

This paper presents a compact RF energy harvesting wireless sensor node with the antenna, rectifier, energy management circuits, and load integrated on a single printed circuit board and a total size of 53 mm × 59.77 mm × 4.5 mm. By etching rectangular slots in the radiation patch, the antenna area is reduced by 13.9%. The antenna is tested to have an S11 of −24.9 dB at 2.437 GHz and a maximum gain of 4.8 dBi. The rectifier has a maximum RF-to-DC conversion efficiency of 52.53% at 7 dBm input energy. The proposed WSN can achieve self-powered operation at a distance of 13.4 m from the transmitter source. To enhance the conversion efficiency under different input energy densities, this paper establishes an energy model for two operating modes and proposes an energy-intensity adaptive management algorithm. The experiments demonstrated that the proposed WSN can effectively distinguish between the two operating modes based on input energy intensity and realize efficient energy management.

## 1. Introduction

With the rapid development of the Internet of Things and communication technology, wireless sensor nodes (WSNs) have been widely applied. Traditional wireless sensor nodes are restricted by the requirements of volume, weight, and battery capacity. As a remedy, wireless sensor nodes based on ambient energy sources, including vibration, photovoltaic, thermoelectric, and radio frequency (RF), have emerged [1,2,3,4,5]. RF energy sources, such as wireless routers and mobile base stations, are widely distributed in the environment and emit electromagnetic waves (RF energy) all the time. RF energy has become a research hotspot in self-powered WSN research due to its long-distance wireless transmission, wide distribution, and resistance to obstruction and environmental influences [6,7,8].

An RF energy harvesting WSN (RF-EH WSN) typically consists of an RF energy harvester and its loads, including a microcontroller, wireless transceiver sensors, etc. An RF energy harvester generally consists of an energy-receiving antenna, a rectifier for RF to DC conversion, and energy management circuits. Generally, the rectifier contains an impedance-matching network to achieve impedance matching with the antenna and maximize the efficiency of energy transmission. The antenna and rectifier, referred to as rectenna, are the core devices for RF energy harvesting. A compact printed rectenna is important for reducing the total size and flexible integration in an electronic system [9,10]. In some existing designs, the antenna and the rectifier are not fabricated on a printed circuit board (PCB) but are connected through an SMA interface, resulting in a large overall area [11,12,13]. The antenna design and rectifier design originally used different software programs. Through separate design and the use of SMA interfaces for lossless connections, it is possible to minimize interference between the antenna and rectifier. But, this will undoubtedly result in a complex and challenging-to-integrate system architecture. For example, in reference [11], the antenna size is 50 × 50 mm, and the rectifier size is 21.7 mm × 107.6 mm. These two sizes cannot achieve effective integration. Some rectennas are printed on a board, but the integration of rectifier antennas can lead to a decrease in antenna gain due to changes in antenna parameters. In the referenced literature [14,15], a reflector was used to enhance antenna gain, and this resulted in a significant increase in overall dimensions or thickness. Rare studies have achieved the integration of antennas and rectifiers with almost no increase in antenna size and without affecting antenna performance.

According to the Friis formula [16], the transmission loss of RF energy is significantly influenced by the distance between the source and the receiver. Additionally, due to spatial wave reflections, the RF energy density reaching the wireless sensor network (WSN) exhibits significant variations at different locations. Based on the previous research, there are two main operating modes for RF-EH WSNs. Mode A involves continuous data acquisition and transmission until the energy storage components are depleted [7,8,17,18,19,20,21]. Mode B involves periodic sleep, wake-up, data acquisition, and transmission [22,23,24,25,26,27,28].

Mode B requires the input energy to exceed the system’s sleep power consumption in order to enable charging during the sleep period. This imposes more strict demands on the intensity of the input RF energy. Mode A incurs no standby power consumption during the two consecutive operating intervals, resulting in a faster charging rate. Compared to mode A, mode B only requires a single power-up to sustain continuous operation, reducing power-up losses. However, it introduces additional standby power consumption. It is evident that mode A is capable of accommodating lower input energy intensities. As the input energy increases, mode B gradually becomes more advantageous. When the input energy is sufficiently high for direct driving of the system, there is a negligible difference between the two methods. To the best of our knowledge, previous research primarily employed either mode B or mode A without selecting the working mode based on input energy intensity.

To address the above-mentioned issues, a compact RF-EH WSN with an antenna, a rectifier, and a load designed on a single PCB is proposed in this paper. A compact microstrip antenna structure is proposed. Lumped component matching and the rectifier are used instead of the 50 Ω feedline, which does not introduce additional area and has a relatively minor impact on antenna performance. The integrated RF-EH WSN adds only a small amount of area compared to the antenna. Additionally, an energy mathematical model for two operating modes of the RF-EH WSN is established, and an adaptive energy management algorithm is proposed. This algorithm enables the selection of the most suitable operating mode based on the input energy intensity, aiming to maximize the utilization of RF energy. The contributions of this paper are concluded as follows.

(1)A compact RF energy harvesting wireless sensor node with a total size of 53 mm × 59.77 mm × 4.5 mm;(2)An energy mathematical model for two operating modes of the proposed RF-EH WSN;(3)An adaptive energy management algorithm that enables the selection of the most suitable operating mode based on the input energy intensity.

The remainder of this paper is organized as follows: Section 2 presents a comprehensive description of the complete design of the proposed RF-EH WSN. Section 3 provides a detailed introduction to the proposed energy model and energy management algorithm. Section 4 discusses the results obtained, as well as crucial design considerations. Section 5 concludes the paper.

## 2. Design of the RF-Powered Wireless Sensor Node

### 2.1. Design of the Antenna

The structure and photograph of the proposed antenna are shown in Figure 1a,b, respectively. The antenna was simulated in a High-Frequency Structure Simulator (HFSS) and fabricated on a Polytetrafluoroethylene (PTFE) substrate of 1.52 mm thickness. The antenna utilizes an enhanced side-fed microstrip antenna structure. The dimensional parameters of the antenna and the rectifier are shown in Table 1. The antenna achieves impedance matching by varying the feed depth L2 and slotted width W2. In Figure 1a, the current transmission paths are extended, and the antenna area is reduced by etching three rectangular slots with a width of W3 and a length of L3. Changing the width and length of the slots can adjust the current distribution within the radiation patch, thus modifying the antenna’s resonance frequency and dimension. The comparison of the current distribution before and after slotting is shown in Figure 1c,d. It is evident that the current density and transmission path are significantly increased and extended after slotting. The total area of the antenna shown in Figure 1c is 56 mm × 47.7 mm. Compared with the proposed antenna (53 mm × 43.4 mm), the total area decreases by 13.9% after slotting.

Figure 2a shows the comparison between the simulated and measured S11 of the proposed antenna. The S11 test was performed with an Agilent vector network analyzer, model N9916A. The measured center frequency of the proposed antenna is located at 2.437 GHz with an S11 of −24.9 dBi. Figure 2b shows the simulations of the radiation characteristics. The proposed antenna has an omnidirectional radiation efficiency of 89.11% and a maximum gain of 4.804 dBi, which can be effectively used for RF energy harvesting.

### 2.2. Design of the Rectifier

The rectifier is designed in the Advanced Design System (ADS). The rectifier utilizes the unidirectional conduction of diodes for rectification and combines it with capacitors to achieve multi-stage voltage boosting. In the design process, it is necessary to add an impedance-matching network at the front end of the rectifier to efficiently transform the antenna impedance to the rectifier’s input impedance. Due to the nonlinearity of diodes, the input impedance of the rectifier varies with load resistance and input power. Therefore, it is necessary to perform input power and load pull simulations in ADS to achieve the optimal design. The rectifier and the antenna are fabricated on a single PTFE substrate. After integration, the rectifying antenna is of the same size as the antenna. As shown in Figure 3c, the proposed microstrip antenna has a central feed line of 50 Ω impedance, which is led from the radiating patch and matched to the rectifier by lumped components (a 6.8 nH and a 7.5 nH inductor from Murata). The isolation capacitor used is 5.1 pF from Murata, and the diode is HSMS2852 from Broadcom.

The RF-to-DC conversion efficiency is shown in Figure 4, which was calculated according to Equation (1), where $P_{out}$, $P_{in}$, $V_{out}$, and R denote the output energy, input energy, output voltage, and the load resistance of the rectifier. During testing, the RF signal source used was the DSG836 from RIGOL, and the voltmeter used was the 15B+ from FLUKE. In Figure 4, the RF-to-DC conversion efficiencies are higher than 30% for $P_{in}$ between −5 dBm and 11 dBm and reach a peak of 52.53% when $P_{in}$ is 7 dBm. Compared to references [29] (maximum efficiency 37%) and [30] (maximum efficiency 40.29%), this paper has a more significant advantage.
(1)$$\eta =\frac{P_{out}}{P_{in}}=\frac{V_{out}{}^{2}}{R\cdot P_{in}}$$

### 2.3. System Integration

The rectifying antenna is integrated with the energy management circuits and the load to create the RF-EH WSN. The proposed RF-EH WSN and its internal connection relationship are shown in Figure 5a and b, respectively. The total size is 53 mm × 59.77 mm × 4.5 mm. The energy management circuit consists of a load switch chip, a capacitor, a BUCK chip, and a power management integrated chip (PMIC) that supports a minimum cold start voltage of 0.33 V and a minimum input power of 15 uW. The PMIC can perform threshold management and control the on/off state of the downstream load switch. In this paper, the high and low voltage thresholds are set to 5.15 V and 2.8 V, respectively. When the voltage across the energy storage capacitor exceeds 5.15 V, the PMIC controls the load switch to turn on, allowing the capacitor to discharge. When the voltage of the energy storage capacitor drops below 2.8 V, the PMIC controls the load switch to turn off, stopping the discharge of the capacitor. The PMIC outputs a voltage of 2.8 V through BUCK conversion. The load of the BUCK includes an ultra-low-power analog–digital converter (ADC), a low-power microcontroller, a sensor, and a wireless transceiver. The ADC is used to measure the capacitor voltage for implementing the energy management algorithm in the next section. After the algorithm finishes running, the ADC is powered off using a load switch. Table 2 lists the chip models and manufacturers used in the WSN.

In paper [31], an analysis was conducted on the charge consumption of the load used in this paper at an MCU main frequency of 4 MHz. The charge consumption for power-up and single data acquisition and transmission are 55 µC (referred to as $Q_{{}_{init}}$) and 43.41 µC (referred to as $Q_{Single}$), respectively. The total charge consumption for a single operation cycle is 98.41 µC. To ensure the reliable operation of the system, $Q_{Init}+3\times Q_{Single}$ is used as the load charge amount and combined with Equation (2) to estimate the energy-storage capacitance. In Equation (2), C represents the capacitance of the energy storage capacitor, $U_{H}$ denotes the high discharge threshold of the PMIC (5.15 V), $U_{L}$ denotes the low discharge threshold of the PMIC (2.8 V), $\eta_{B}$ denotes the BUCK conversion efficiency (estimated as 90%), $P_{LOAD}$ denotes the load power consumption, and $U_{B}$ indicates the output voltage of BUCK (2.8 V). The estimated capacitance value is 72.03 uF, and a capacitance value of 100 uF is selected. It is evident that in the “store-and-release” energy management mode, to ensure the system’s stable power-up and operation, a significant amount of redundant power consumption is present in addition to the consumption of single data acquisition and transmission.
(2)$$\frac{1}{2}C\left(U_{H}{}^{2}-U_{L}{}^{2}\right)\times \eta_{B}=\int_{0}^{T}P_{LOAD}dt=U_{B}\times \left(Q_{Init}+3\times Q_{Single}\right)$$

## 3. Energy Intensity Adaptive Management Algorithm

In this paper, two operating modes for RF-EH WSN are defined. For the same input power of the PMIC, the mode with the shorter single operating cycle consumes less energy and has higher efficiency. For mode A, the charging rate is higher during its charging process, as there is no load sleep power consumption, but the required total charging amount increases as it runs out of power in a single operation. For mode B, since it does not require secondary initialization, the capacitor voltage does not drop to the low-voltage threshold after each acquisition and sending operation. The capacitor is always in the charging cycle from the intermediate voltage charging to the high voltage threshold. There is a load sleep power consumption during the charging process of mode B, for which the charging rate is slower compared to mode A.

This paper conducts a power consumption comparative analysis between the two modes and proposes an intelligent energy management algorithm. The algorithm adaptively adjusts the system operating mode according to the RF energy input intensity and calculates the system working cycle under mode B to maximize the energy utilization efficiency. Table 3 lists the variable descriptions and values used in the estimation.

Equation (3) demonstrates the relationship between the output power of the PMIC ($P_{Pout}$), the capacitance (C), the capacitor voltage (U), and the capacitor self-leakage current ($I_{s1}$) during the charging process in mode A. The leakage current of the load switch (2 nA) is relatively small, which is not considered in Equation (3). To simplify the derivation, the second term in Equation (3) is treated as a constant, as shown in Equation (4), where $U_{H}$ and $U_{L}$ denote the high and low threshold voltage of the PMIC. Bringing Equation (4) into Equation (3), simplifying and integrating at both ends yields Equation (5), where $T_{A-charge}$ denotes the charging time in mode A. After simplification, $T_{A-charge}$ can be expressed by Equation (6). Since the capacitor selection is derating and the PMIC keeps supplying energy during the discharge process, the energy storage capacitor can supply the system to achieve multiple data acquisition and transmission work. The total duration of N times of acquisition and transmission in mode A is expressed by Equation (7), where $t_{Single}$ denotes the time of single acquisition and transmission, and $t_{init}$ denotes the system-initialization time. N is calculated by Equation (8), which expresses the load power dissipation in relation to $P_{Pout}$ and the voltage drop of the energy storage capacitor in a complete discharge. After simplification, N can be expressed by Equation (9).
(3)$$d\left(\frac{1}{2}\times C\left(U^{2}\right)\right)+I_{s1}\times U\times dt=P_{Pout}\times dt$$
(4)$$I_{s1}\times U=I_{s1}\times \frac{U_{H}+U_{L}}{2}$$
(5)$$\int_{U_{L}}^{U_{H}}\frac{U\times dU}{P_{Pout}-I_{s1}\times \frac{U_{H}+U_{L}}{2}}=\int_{0}^{T_{A-charge}}\frac{dt}{C}$$
(6)$$T_{A-charge}=\frac{1}{P_{Pout}-I_{s1}\times \frac{U_{H}+U_{L}}{2}}\left(\frac{1}{2}\times C\times (U_{H}^{2}-U_{L}^{2})\right)$$
(7)$$T_{A}=T_{A-charge}+T_{A-discharge}=T_{A-charge}+t_{init}+N\times t_{Single}$$
(8)$$\frac{1}{2}\times C\times (U_{H}^{2}-U_{L}^{2})+P_{Pout}\times (N\times t_{Single}+t_{init})=\frac{\left(N\times Q_{Single}+Q_{init}\right)\times U_{B}}{\eta_{B}}$$
(9)$$N=\frac{\frac{Q_{init}\times U_{B}}{\eta_{B}}-P_{Pout}\times t_{init}-\frac{1}{2}\times C\times (U_{H}^{2}-U_{L}^{2})}{P_{Pout}\times t_{Single}-\frac{Q_{Single}\times U_{B}}{\eta_{B}}}$$

Equation (10) demonstrates the relationship between the output power of PMIC ($P_{Pout}$) and its composition during the charging process in mode B, including the load sleeping consumption (the first term), the capacitor self-leakage consumption (the second term), and the capacitor charging consumption (the third term). $I_{s2}$ is the load sleeping current. To simplify the calculation, the second term is simplified by Equation (11), where $U_{1}$ and $U_{2}$ are the voltages across the capacitor after initialization and single data acquisition and transmission, respectively. After power-on initialization, the capacitor voltage continues to be maintained between $U_{1}$ and $U_{2}$. Equation (11) is brought into Equation (10) and simplified and integrated at both ends to obtain Equation (12), where $T_{{}_{B-charge}}$ is the charging time in mode B. Since the capacitor voltage charges to the high threshold, drops to $U_{1}$ after initialization, and then no longer drops to the low threshold, the estimation of the operating cycle does not include the charging time from the low voltage threshold to the high voltage threshold but only the charging time from $U_{2}$ to $U_{1}$. The energy consumption of power-up initialization and each data acquisition and transmission are expressed by Equation (13) and Equation (14), respectively. By substituting the derived values of $U_{1}$ and $U_{2}$ in Equations (13) and (14) into Equation (12), the charging time in mode B ($T_{B-charge}$) can be determined, as shown in Equation (15). Equation (16) represents the total duration of N times of data acquisition and transmission in mode B, where $T_{{}_{B-charge}}$ is the time consumed per charge and $t_{Single}$ is the time consumed per discharge.
(10)$$\frac{I_{s2}\times U_{B}}{\eta_{B}}+I_{s1}\times U+\frac{d\left(\frac{1}{2}\times C\left(U^{2}\right)\right)}{dt}=P_{Pout}$$
(11)$$I_{s1}\times U=I_{s1}\times \frac{U_{1}+U_{2}}{2}$$
(12)$$\int_{U_{2}}^{U_{1}}\frac{CUdU}{P_{Pout}-\frac{I_{s2}\times U_{B}}{\eta_{B}}-I_{s1}\times \frac{U_{1}+U_{2}}{2}}=\int_{0}^{T_{B-charge}}dt$$
(13)$$\frac{1}{2}\times C\times (U_{H}^{2}-U_{1}^{2})+P_{Pout}\times t_{init}=\frac{Q_{init}\times U_{B}}{\eta_{B}}$$
(14)$$\frac{1}{2}\times C\times (U_{1}^{2}-U_{2}^{2})+P_{Pout}\times t_{Single}=\frac{Q_{Single}\times U_{B}}{\eta_{B}}$$
(15)$$T_{B-charge}=\frac{\frac{Q_{Single}\times U_{B}}{\eta_{B}}-P_{Pout}\times t_{Single}}{P_{Pout}-\frac{I_{s2}\times U_{B}}{\eta_{B}}-I_{s1}\times \frac{U_{1}+U_{2}}{2}}$$
(16)$$T_{B}=N\times T_{B-charge}+N\times t_{Single}$$

The time difference between the two operating modes with the same N is shown in Equation (17). As previously mentioned, N is the number of data collection and transmission cycles supported by a single discharge in mode A, and mode B applies the same N for comparison with mode A. The input RF energy density and $P_{Pout}$ are unknown for the proposed WSN, but they can be estimated by sampling the capacitor voltage using an ADC. After power-on initialization, $U_{1}$ can be measured by the ADC and brought into Equation (13) and Equation (14) to calculate $P_{Pout}$ and $U_{2}$, respectively. Using Equation (17), the operating time difference deltaT between the two operating modes can finally be calculated to determine which operating mode to choose.
(17)$$\begin{array}{l} deltaT=T_{A}-T_{B}=T_{A-charge}+t_{init}+N\times t_{Single}-\left(N\times T_{B-charge}+N\times t_{Single}\right)=T_{A-charge}-N\times T_{B-charge}+t_{init}\\ =\frac{\frac{1}{2}\;\times \;C\;\times \;(U_{H}^{2}-U_{L}^{2})}{P_{Pout}-I_{s1}\;\times \;\frac{U_{H}+U_{L}}{2}}-N\;\times \;\frac{\frac{Q_{Single}\;\times \;U_{B}}{\eta_{B}}-P_{Pout}\;\times \;t_{Single}}{P_{Pout}-\frac{I_{s2}\;\times \;U_{B}}{\eta_{B}}-I_{s1}\;\times \;\frac{U_{1}+U_{2}}{2}}+t_{init} \end{array}$$

Figure 6a shows the simulation of $T_{A-charge}$, $N\times T_{B-charge}$, deltaT, and N versus $U_{1}$. Figure 6b shows the simulation of $P_{Pout}$ versus $U_{1}$. The system uses a 16-bit ultra-low power ADC, which can achieve a sampling accuracy of less than 0.1 mV. The range of $U_{1}$ in Figure 6 is set to exceed 4.8066 V because when it falls below this voltage, $P_{Pout}$ will approach the load sleeping consumption, resulting in an excessively long calculated sleeping time, and only mode A can be selected.

$P_{Pout}$ is directly proportional to $U_{1}$, as observed in Figure 6b. In Figure 6a, mode B consumes much more time than mode A when $P_{Pout}$ is relatively low. When $U_{1}$ exceeds 4.8077 V and $P_{Pout}$ exceeds 62 μW, two operating modes consume an equal time of 15.62 s. At this point, N is about 6; that is, the single operating time of mode B is about 1/6 of mode A. As $P_{Pout}$ increases, deltaT rises and then falls, reaching a maximum value of 0.782 s when $U_{1}$ equals 4.809 V and $P_{Pout}$ is 115 μW, and then gradually decreases. As the energy input rises, the charging time of the two operating modes is gradually shortened, and the discharging time is gradually extended. When the input energy is high enough, the charging time will occupy a very small proportion, and even after the charging is completed, the power will no longer be discharged, at which time there is no significant difference between the two operating modes. For example, when $U_{1}$ > 4.8765 V and $P_{Pout}$ exceeds 2.97 mW, the numerator term in Equation (15) will tend to zero, i.e., the energy consumption in the discharge phase will be completely supplemented by $P_{Pout}$, and the cycle calculation will be meaningless at this point.

In the time consumption comparison of the two operating modes, this article has assumed no communication quality and calibration retransmission problems. However, in practical application scenarios, the charge consumed by a single acquisition and transmission is usually higher than $Q_{Single}$ due to communication distance, environmental electromagnetic interference, and other factors, and the estimation for N will no longer be accurate. The previous projections in this article are used to quantitatively describe the energy and time consumption differences between the two operating modes. In the actual system operating algorithm, the sleep charging cycle of mode B can be appropriately increased to cope with the sudden additional communication power consumption.

Based on the above analysis, an intelligent energy intensity adaptive management algorithm is proposed, as shown in Figure 7. The algorithm realizes fine-grained management of energy by three energy intensity divisions, achieving the highest efficiency of data acquisition and transmission under the same energy intensity input and improving data transmission reliability.

In this algorithm, mode A is used when the energy input intensity is relatively low (≤62 μW) or high (≥2.97 mW), and mode B is used when the energy intensity is intermediate. In Figure 7, after the system power-up initialization is completed, the voltage $U_{1}$ across the capacitor is measured by the ADC and the RF input energy intensity is judged with Equation (18) (condition 1), where 4.8077 V is used to determine whether the input energy intensity is too low (≤62 μW) to sustain sleep power consumption, and 4.8765 V is used to determine whether the energy is sufficiently high (≥2.97 mW) to eliminate the need for sleep. If condition 1 is not satisfied, the system works according to mode A. If condition 1 is satisfied, the system works according to working mode B, sleeping and waking up according to the calculated cycle. The periodic calculation is shown in Equation (19), obtained by substituting the $U_{1}$ expression of $P_{Pout}$ derived from Equation (13) into Equation (15) and multiplying by 1.5. By simplifying the calculation results in advance in Equation (19), the amount of microcontroller calculation and calculation power consumption can be greatly reduced. The charging duration is appropriately increased by multiplying it by 1.5 times to cope with sudden additional communication power consumption.
(18)$$4.8077\;\;V<U_{1}<4.8765\;\;V$$
(19)$$T_{B-charge}=-0.0678+\frac{1}{21.90669\times U_{1}^{2}-506.11054}$$

## 4. Validation

Figure 8 displays the test results of the proposed energy management algorithm. The testing oscilloscope is SDS2504X from SIGLENT. The RF signal source used is the DSG836 from RIGOL. The transmitting antenna is the proposed antenna in this paper. The RF signal source transmitted energy at different power levels, and then the energy was received by the RF-EH WSN. An oscilloscope was used to monitor the voltage across the energy storage capacitor. In Figure 8a, due to the weak energy input, the voltage measured after the power-up initialization is less than or equal to 4.8077 V, so the system operates according to mode A. At this energy input intensity, the duration required for charging the capacitor exceeds 15.62 s. Note that the two charging durations during this interval are not equal. This phenomenon is attributed to the over-discharge of the capacitor, which results from the untimely closing of the load switch during the wireless transmitting phase of the discharge process. This issue is influenced by the random operating state and cannot be accounted for in theoretical estimations. In the case of operating mode B, the aforementioned issue does not arise as the load switch remains continuously activated.

For Figure 8b, the energy storage capacitor discharges after the voltage reaches 5.15 V, and the capacitor voltage is measured to be 4.8425 V, which triggers the MCU to operate in operating mode B with a cycle time of 63.83 ms. While the system is performing the cycle calculation, the energy storage capacitor voltage decreases, after which the first data acquisition and transmission is performed. During each data acquisition and sending process, there is a waiting time of 26 ms for sensor data conversion. In this period, the system is in a sleeping state with low power consumption, and the storage capacitor voltage rises until the MCU wakes up. After the data acquisition is completed, the MCU sends out the data and the capacitor voltage drops. Then, the system enters the sleeping state, and the capacitor voltage starts to rise. In the second half of Figure 8b, there is a burst of additional energy consumption, which is mainly caused by the retransmission of the transceiver due to the instability of the wireless communication. When an additional power consumption is generated, the capacitor will ramp up at a higher rate at low voltages due to the fixed duration of the single charge. In addition, there is a margin in the calculated charging cycle, so the capacitor voltage will gradually return to the original level position over time. In Figure 7 and Figure 8b, the sleep time is set again after the work mode B acquisition and sending is completed, which can ensure that the charging time is fixed each time to prevent the effect of irregular discharge time on charging. In Figure 8c, after power-up, the system will send data acquisition continuously according to operating mode A. Due to the high input energy, the voltage of the energy storage capacitor is almost constant. The validation results demonstrate the efficacy of the proposed intelligent energy management algorithm in distinguishing between different energy intensities and selecting the appropriate operating mode to improve energy utilization efficiency.

An RF energy-harvesting experiment was conducted, as illustrated in Figure 9. A 27 dBm signal source and a 14 dBi panel antenna were employed as the RF power source. The voltage across the WSN’s capacitor was measured using a multimeter (FLUKE 15B+). A host node was connected to a laptop to receive the sensor data. The experiments demonstrated that the proposed WSN can achieve self-powered operation at a distance of 13.4 m from the transmitter source. This distance can be influenced by the environment (such as reflections from the ground, walls, etc.).

## 5. Discussion

This paper introduces a compact RF-EH WSN. Table 4 presents a comparison between this paper and recent studies. In previous studies, commercially available antennas were widely employed, and the systems were not integrated on the same PCB board, resulting in challenges in quantifying the system volume [23,29,32,33]. By comparing with the existing literature, it is evident that the proposed RF-EH WSN exhibits significant advantages in terms of integration and operating range, and it possesses the unique capability of energy intensity adaptive management.

This paper achieved a self-powered distance of up to 13.4 m, which significantly surpasses previous designs. There are two reasons for this. First, this paper made certain design enhancements to improve antenna gain and rectifier conversion efficiency. Second, the load in this paper has been optimized and designed to have an extremely low single-operation charge consumption (98.41 µC) [30]. This reduction in energy storage capacitor value and self-leakage current contributed to a decrease in the minimum required input energy. Both factors significantly extended the self-powered distance.

The energy mathematical model and energy management algorithm proposed in this paper are simple and highly portable. When applying this energy management model to new WSNs, we only need to measure the variable values in Table 3 and plug them into Equations (15) and (17) to calculate the boundary voltage of $U_{1}$. Then, based on the algorithm shown in Figure 7 and the just-computed boundary voltage, the WSNs can switch operating modes and calculate the sleep duration to achieve optimal energy management.

## 6. Conclusions

A compact RF-EH WSN is proposed with a total size of 53 mm × 59.77 mm × 4.5 mm. By printing the total RF-EH WSN on a single PCB, a high level of integration is achieved. The antenna area is further reduced by etching rectangular slots. An adaptive RF-energy-intensity management algorithm is proposed. Through the derivation of energy models for two common operating modes, the boundary conditions for mode switching are established. The derivation and application of this energy management algorithm have broad adaptability. The proposed RF-EH WSN can achieve self-powered operation at a distance of 13.4 m from the RF signal source, providing a reference for powering wireless sensor nodes in industries, military applications, and other fields. In order to achieve a compact design, lumped element matching was used in this paper, resulting in a maximum RF-to-DC conversion efficiency of only 52.53% at 7dBm input power. In the future, more efficient energy harvesting and a longer self-powered distance can be achieved by changing the matching method and antenna structure.

## Figures and Tables

**Figure 1 sensors-23-08641-f001:**
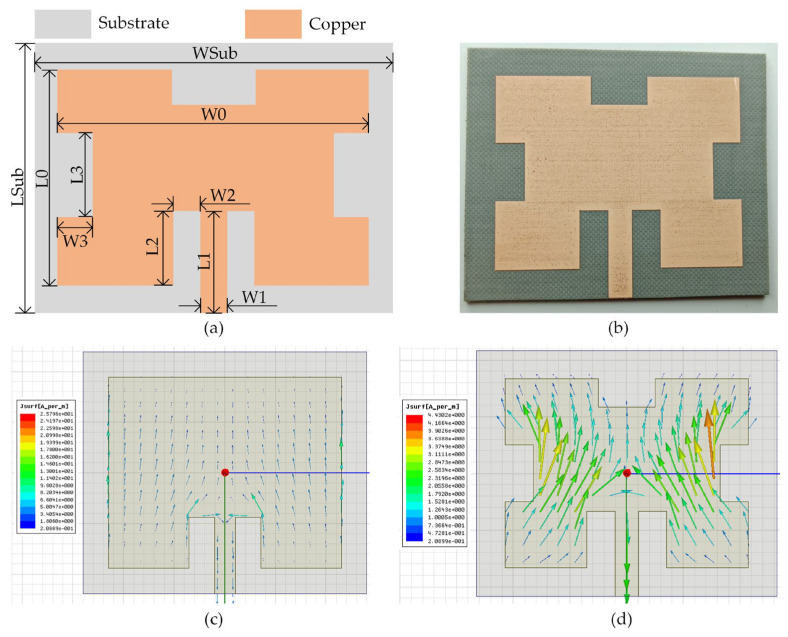
Antenna. (**a**) Structure of top layer; (**b**) photograph of the top layer; (**c**) current distribution without rectangular slots; and (**d**) current distribution with rectangular slots.

**Figure 2 sensors-23-08641-f002:**
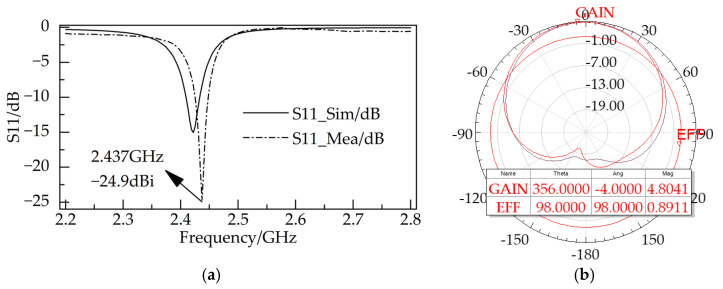
(**a**) Simulated and measured S11; (**b**) simulated radiation pattern.

**Figure 3 sensors-23-08641-f003:**
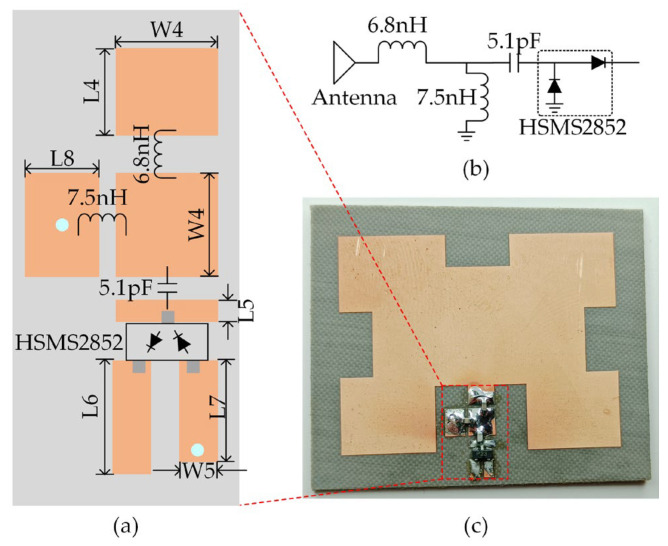
(**a**) Structure and dimension of the rectifier; (**b**) circuit diagram of the rectifier; and (**c**) photograph of the rectenna.

**Figure 4 sensors-23-08641-f004:**
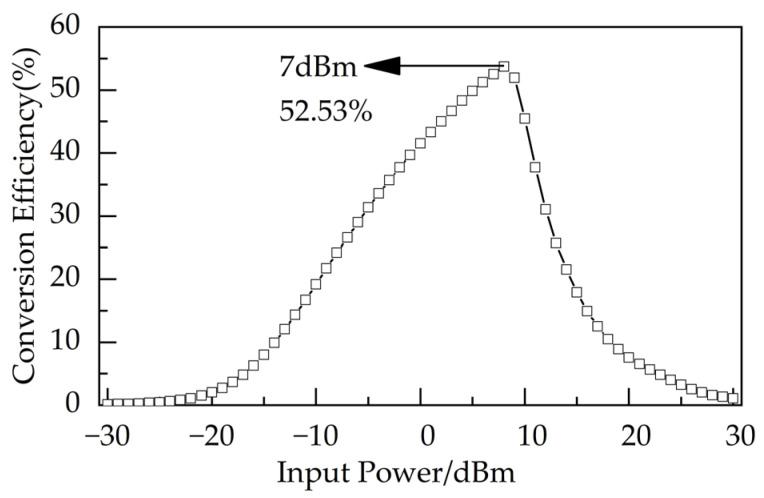
RF-to-DC conversion efficiency of the proposed rectifier.

**Figure 5 sensors-23-08641-f005:**
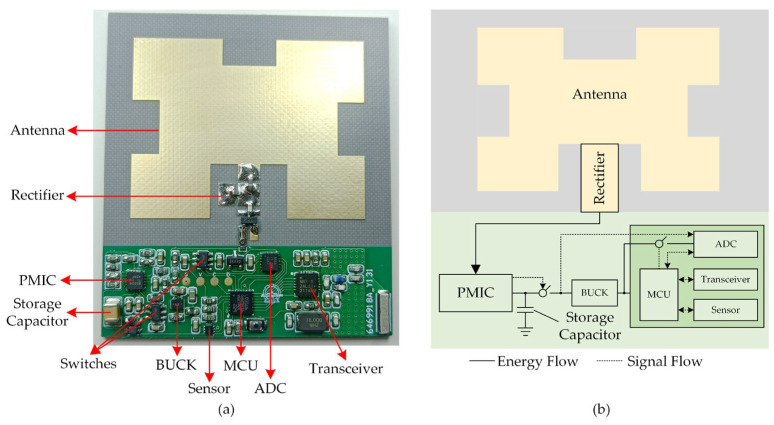
(**a**) Proposed RF-EH WSN; (**b**) internal connection relationship of the proposed WSN.

**Figure 6 sensors-23-08641-f006:**
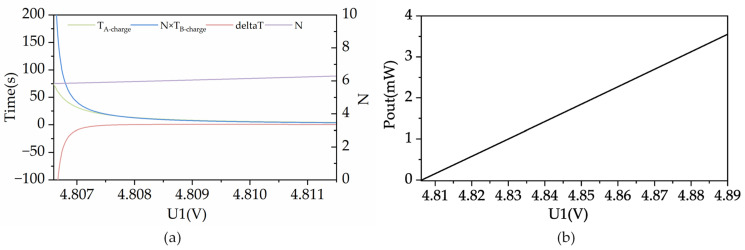
(**a**) Simulation of $T_{A-charge}$, $N\times T_{B-charge}$, deltaT, and N versus $U_{1}$; (**b**) simulation of $P_{Pout}$ versus $U_{1}$.

**Figure 7 sensors-23-08641-f007:**
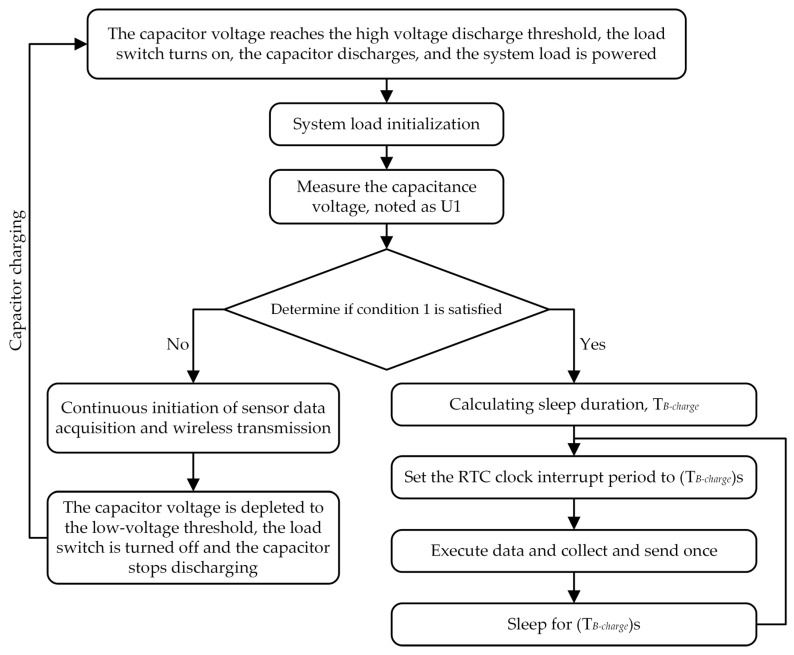
Proposed energy intensity adaptive management algorithm.

**Figure 8 sensors-23-08641-f008:**
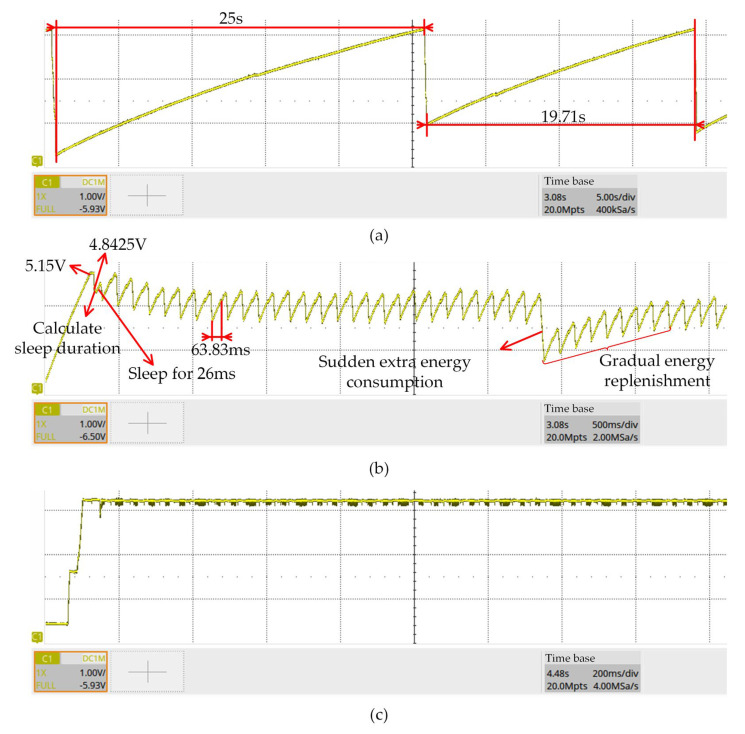
Test results of the proposed energy management algorithm. (**a**) Low energy input intensity, with the system operating in mode A. The time base is 5 s/div. (**b**) Intermediate energy input intensity, with the system operating in mode B. The time base is 500 ms/div. (**c**) High energy input intensity, with the system operating in mode A. The time base is 200 ms/div.

**Figure 9 sensors-23-08641-f009:**
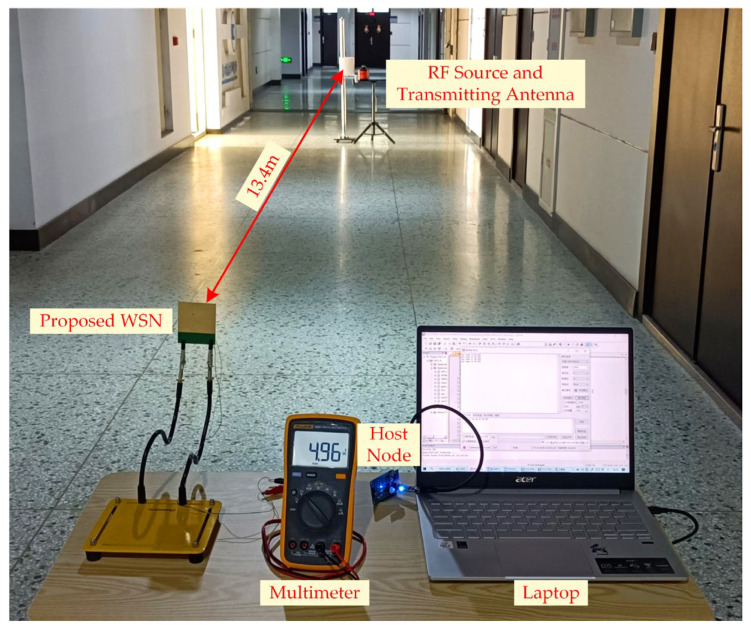
RF energy-harvesting experiment of the proposed RF-EH WSN.

**Table 1 sensors-23-08641-t001:** Dimensional parameters of antenna and rectifier.

Parameter of Antenna	W0	L0	W1	L1	W2	L2	W3	L3	Wsub	Lsub
Value (mm)	43	33.4	4.15	15	5	10	5	10	53	43.4
Parameter of Rectifier	W4	L4	L5	L6	L7	L8	W5			
Value (mm)	4.15	3	1	2.75	3	3.08	1.5			

**Table 2 sensors-23-08641-t002:** Chip models and manufacturers used in the WSN.

	PMIC	BUCK	MCU	ADC	Transceiver	Sensor
Model	BQ25504RGTT	TPS62840	STM32L031G6	ADS8866IDRCR	nRF24L01	TMP102AIDRLR
Manufacturer	TI	TI	ST	TI	NORDIC	TI

**Table 3 sensors-23-08641-t003:** The variable descriptions and values used in the estimation.

	Description	Value
C	Capacitance of the storage capacitor	100 μF
$I_{s1}$	Self-leakage current of the storage capacitor	260 nA
$U_{H}$	High threshold voltage of the PMIC	5.15 V
$U_{L}$	Low threshold voltage of the PMIC	2.8 V
$Q_{Init}$	Power-on charge consumption of the load	55 μC
$Q_{Single}$	Single data collection and transmission charge consumption	43.41 μC
$t_{init}$	Power on duration	11.42 ms
$t_{Single}$	Single data collection and transmission duration	45.2 ms
$U_{B}$	Output voltage of BUCK	2.8 V
$\eta_{B}$	Conversion efficiency of BUCK	90%
$I_{s2}$	Current for load in sleep mode	3.5 uA

**Table 4 sensors-23-08641-t004:** Comparison of the proposed WSN with reported RF-EH WSN.

Reference	Total Size	Maximum Gain	Maximum Self-Powered Distance	Adaptive Energy Management
[32]	-	-	2 m	no
[22]	-	-	2.3 m	no
[29]	-	-	1.5 m	no
[33]	125 mm × 140 mm	-	-	no
[15]	150 mm × 90 mm × 50 mm	6.6 dBi	1.5 m	no
[21]	30 mm × 30 mm	4 dBi	1.12 m	no
This paper	53 mm × 59.77 mm × 4.5 mm	4.804 dBi	13.4 m	yes

## Data Availability

Not applicable.

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
