# Peer review of "A Compact RF Energy Harvesting Wireless Sensor Node with an Energy Intensity Adaptive Management Algorithm"

_sensors, 2023, doi:10.3390/s23208641_

Round 1
Reviewer 1 Report
· The abstract should briefly mention the key contributions and findings of the paper.
· Provide more context and motivation for the research in the introduction.
· Clearly state the problem you are addressing and why it is important.
· Include a more detailed description of the antenna design, such as the type of antenna used, dimensions, and materials.
· Explain the significance of reducing the antenna area and its impact on the overall device.
· Provide some insight into the choice of frequency (2.437GHz) and why it was selected.
· Elaborate on the design and working principles of the rectifier.
· Discuss the factors affecting the RF-to-DC conversion efficiency, including any trade-offs made during the design.
· Compare the achieved efficiency with existing works in this area for context.
· Clarify the key components of the wireless sensor node, including details on energy management circuits and load specifications.
· Provide more information on the sensor(s) used and their role in the system.
· Discuss the energy requirements of the sensor node in different operating modes.
· Describe the experimental setup and methodology in more detail, including the equipment used for testing.
· Mention any environmental conditions or variables that may have affected the results.
· Provide statistical measures, such as standard deviations, for the experimental results to indicate the level of confidence in the findings.
· Explain the energy model in greater detail, including the equations and parameters used.
· Describe the algorithm for adaptive energy management, step by step.
· Discuss the algorithm's complexity and any computational limitations.
· Interpret the results more comprehensively, highlighting the practical implications of the findings.
· Discuss any limitations or potential sources of error in the experiments.
· Consider discussing the implications of your work for real-world applications and future research directions.
· Summarize the main contributions of the paper concisely.
· Emphasize how your work advances the state-of-the-art in RF energy harvesting and wireless sensor networks.
· Ensure that all references are cited correctly and consistently throughout the paper.
· Include recent and relevant references to related work in the field.
· Check the figures and tables for clarity and accuracy.
· Ensure that all figures and tables are referenced and explained in the text.
· Proofread the paper for grammar, punctuation, and spelling errors.
· Maintain a consistent writing style and formatting throughout the paper.
· Provide additional information or supplementary materials to aid in the reproducibility of your experiments, if possible.
Author Response
Please see the attachment. The yellow-highlighted text represents the modified sections, while the green-highlighted sections are provided for reference and have not been altered.

Reviewer 2 Report
1. Line 13, 90,334, etc. please put the appropriate symbol × instead of *
2. Introduction part is weak and short.
3. The authors have to mention their contribution in the introduction section.
4. The readers are also interested in finding how your work is similar and different from the existing work. Hence the manuscript needs to be revised, keeping this in mind. Try to add details about the similar existing work, which I find is lacking in the manuscript.
5. The author needs to explain the algorithmic complexity of their proposed Energy Intensity Adaptive Management Algorithm.
6. Give the limitations and assumptions of your work.
7. The conclusion part is concise. Try to quantify the result in the conclusion section and add future directions of your proposed work for the new researchers.
Some moderate changes required.
Author Response

(The authors gave the same response as above.)

Reviewer 3 Report
This paper demonstrated a novel design, integration and application of RF -EH WSN. In addition, the author also introduced an energy management algorithm that can adaptively adjusts the system operating mode and maximize energy efficiency.
This paper is well written, and their approach is sound.
Author Response
Thank you very much for taking the time to review this manuscript.
Reviewer 4 Report
*The number of literature references should be increased and references should be analysed in detail.
*Table 2 shows the energy conversion efficiencies of the references compared. Conversion cycle times should also be given.
*The maximum self-power distance has been increased from 2.3m to 13.4m according to the references given. The reason for this should be given in detail in the discussion section.
*In addition, the causes of energy conversion losses should also be explained.
Author Response

(The authors gave the same response as above.)

Round 2
Reviewer 1 Report
Authors have made all the changes as suggested. Hence paper is accepted in its present form.
Reviewer 4 Report
*The authors made the requested revisions in the article.